# Deep Levels and Electron Paramagnetic Resonance Parameters of Substitutional Nitrogen in Silicon from First Principles

**DOI:** 10.3390/nano13142123

**Published:** 2023-07-21

**Authors:** Chloé Simha, Gabriela Herrero-Saboya, Luigi Giacomazzi, Layla Martin-Samos, Anne Hemeryck, Nicolas Richard

**Affiliations:** 1Alternative Energies and Atomic Energy Commission—Military Applications Division—Ile-de-France (CEA-DAM-DIF), Bruyères-Le-Châtel, F-91297 Arpajon, France; nicolas.richard@cea.fr; 2Laboratory for Analysis and Architecture of Systems—National Centre for Scientific Research (LAAS-CNRS), University of Toulouse, CNRS, 7 Avenue du Colonel Roche, F-31400 Toulouse, France; anne.hemeryck@laas.fr; 3National Research Council—Institute Of Materials (CNR-IOM), c/o International School for Advanced Studies (SISSA) Via Bonomea 265, IT-34136 Trieste, Italy; giacomazzi@iom.cnr.it (L.G.); marsamos@iom.cnr.it (L.M.-S.); 4Materials Research Laboratory, University of Nova Gorica, Vipavska 11c, 5270 Ajdovscina, Slovenia

**Keywords:** density functional theory, silicon, defects, electron paramagnetic resonance spectroscopy, deep-level transient spectroscopy

## Abstract

Nitrogen is commonly implanted in silicon to suppress the diffusion of self-interstitials and the formation of voids through the creation of nitrogen–vacancy complexes and nitrogen–nitrogen pairs. Yet, identifying a specific N-related defect via spectroscopic means has proven to be non-trivial. Activation energies obtained from deep-level transient spectroscopy are often assigned to a subset of possible defects that include non-equivalent atomic structures, such as the substitutional nitrogen and the nitrogen–vacancy complex. Paramagnetic N-related defects were the object of several electron paramagnetic spectroscopy investigations which assigned the so-called SL5 signal to the presence of substitutional nitrogen (N${}_{\mathrm{Si}}$). Nevertheless, its behaviour at finite temperatures has been imprecisely linked to the metastability of the N${}_{\mathrm{Si}}$ center. In this work, we build upon the robust identification of the SL5 signature and we establish a theoretical picture of the substitutional nitrogen. Through an understanding of its symmetry-breaking mechanism, we provide a model of its fundamental physical properties (e.g., its energy landscape) based on ab initio calculations. Moreover by including more refined density functional theory-based approaches, we calculate EPR parameters ($\underset{g}{↔}$ and $\underset{A}{↔}$ tensors), elucidating the debate on the metastability of N${}_{\mathrm{Si}}$. Finally, by computing thermodynamic charge transition levels within the GW method, we present reference values for the donor and acceptor levels of N${}_{\mathrm{Si}}$.

## 1. Introduction

Nitrogen is introduced in silicon to enhance its mechanical strength [1,2], augment the precipitation of oxygen [3], suppress the diffusion of self-interstitials [3], anchor dislocations [4] and restrain the formation of voids [5]. Even if for most of these applications the precise role of nitrogen is unclear, the suppression of the formation of voids has been directly correlated to the creation of nitrogen–vacancy complexes [6]. Indeed, several nitrogen-related defects (nitrogen–nitrogen pairs, nitrogen–vacancy complexes, etc.) have been associated with the electrical and/or optical signature of N-doped silicon [7,8,9,10,11,12,13,14,15,16]. However, it has been challenging to assign a spectroscopic signal to a specific defect. In the case of the deep-level transient spectroscopy (DLTS), one set of deep levels is often assigned to the overall presence of nitrogen or to a subset of N-related defects. This ambiguous assignment can even include defects that do not share a similar atomic structure, such as susbtitutional nitrogen or the nitrogen–vacancy complex, for instance [17]. On the other hand, from electron paramagnetic resonance (EPR) spectroscopy several signals have been associated with the presence of nitrogen [18,19]. Among these EPR signals, only the so-called SL5 center has been unequivocally associated with the substitutional nitrogen (N${}_{\mathrm{Si}}$) [18,19,20,21]. In this work, we build upon this solid correlation and we establish a theoretical reference for the spectroscopic signals of substitutional nitrogen in silicon.

From the first EPR measurements [18,19], it was inferred that contrary to other group V substitutional impurities in silicon (e.g., phosphorous and arsenic), nitrogen does not remain in its tetra-coordinated on-center ($T_{d}$) site (Figure 1). Instead, it resides in a tri-coordinated off-center configuration along one of the 〈111〉 crystal directions, with a lower $C_{3v}$ symmetry. This symmetry-breaking (SB) mechanism was first associated with a Jahn–Teller effect [18,19] that predicts the instability of high symmetry configurations with partially occupied degenerate states [22]. This hypothesis was rejected by early electronic structure calculations [23,24] that located the unpaired electron in a single-particle non-degenerate $a_{1}$ state of the $T_{d}$ configuration. The triply degenerate states $t_{2}$ were found within the conduction band and the SB mechanism was referred to as a pseudo Jahn–Teller effect [24].

The energy landscape associated with this pseudo Jahn–Teller mechanism was investigated from electronic structure calculations by displacing the N atom along one of the equivalent 〈111〉 crystal directions [24]. The resulting potential energy surface (PES) presented a small energy barrier of ∼0.05 eV between the on-center and off-center configurations, indicating the possible metastability of the defect (see PES in Figure 1). The authors of this study, however, express caution regarding the accuracy of this value due to the simplicity of their model. Through early EPR stress measurements [18,19], the reorientation between equivalent off-center configurations at finite temperatures (T >35 K) was observed. As a consequence of this thermally activated mechanism, the center was interpretedto be in an average tetrahedral configuration. Later EPR measurements [21,25,26] did, however, associate the existence of an on-center configuration with an increase of the hyperfine splitting when the temperature rises. This evidence was then correlated to the model of the PES in Figure 1 [21,25,26], postulating that the finite temperature dynamics of N${}_{\mathrm{Si}}$ are characterised by the existence of a metastable on-center geometry and the reorientation between equivalent off-center configurations.

Even if the symmetry-breaking mechanism depicted in Figure 1 was undoubtedly associated with the existence of deep levels in the silicon band gap, early optical measurements were incapable of providing a clear picture of their nature and relative position [18,19]. Furthermore, a majority of DLTS activation energies have been attributed to the overall presence of nitrogen in the sample rather than to a specific center [7,8,9,10,11,12,13]. Moreover, previous studies have referred to both the substitutional impurity and the nitrogen–vacancy complex as the NV center, leading to confusion in assigning the observed levels [2,6,17,27]. Among these investigations, two independent set of deep levels were tentatively assigned to substitutional nitrogen: the first set of DLTS activation energies were attributed to a single donor and single acceptor levels at 0.31 eV and 0.08 eV from conduction band, respectively [28], whereas the second DLTS spectra were associated with single and double acceptor levels at 0.64 eV and 0.34 eV from conduction band [17].

On the computational side, recent works have performed state-of-the-art electronic structure calculations on various N-related defects in silicon [29,30,31]. While these studies have addressed the structural properties of substitutional nitrogen [29,30,31], as well as its single-particle band structure [30,31], no attempts were made to explore the possible metastability of the center or estimate its deep levels with a sufficient level of theory. In this work, we perform a thorough investigation of the SB mechanism in substitutional nitrogen from ab initio calculations, enabling us to establish its fundamental physical properties. Moreover, by including more refined DFT-based methods (i.e., linear response treatments and many-body corrections), we are able to establish a direct comparison between the atomistic model of substitutional nitrogen and the spectroscopic signals assigned to it, i.e., EPR parameters and DLTS levels.

## 2. Materials and Methods

Ab initio calculations were performed with the Quantum-ESPRESSO package for electronic structure calculations [32,33]. We used the Perdew–Burke–Ernzerhof (PBE) exchange-correlation functional [34], together with norm-conserving Trouiller–Martins pseudopotentials with GIPAW reconstruction [35,36]. We also employed a plane wave basis set with a kinetic energy cutoff of 84 Ry. A substitutional nitrogen atom is embedded in a silicon cell containing 216 atoms, which is sampled at Γ. For the neutral defect, spin-polarisation is considered. At charge states +1, −1 and −2, a compensating background charge is included.

Defect geometries at different charge states have been optimized, maintaining a constant lattice parameter. The value of this parameter was set to the estimated one for pristine silicon, $a_{\mathrm{PBE}}$ = 5.47 Å. The convergence threshold on forces for the ionic minimisations was set to 0.026 eV/Å. The energy surface is explored by the climbing-NEB method [32,37] as implemented in QE. The convergence threshold of the force orthogonal to the path was fixed to 0.05 eV/Å.

The hyperfine, $\overset{↔}{A}$ and $\overset{↔}{g}$ coupling tensors of the paramagnetic defect were computed using the QE-GIPAW module of the QE package [33,38]. This is achieved through a linear response-based approach, as described in [38,39]. To ensure convergence of these properties [40], the 216-atom supercell was integrated using a 3×3×3 Monkhorst–Pack grid [41,42].

Many-body corrections were computed within the GW method (G${}_{0}$W${}_{0}$ as implemented in the ABINIT code [43,44]). Calculation parameters were kept the same as for QE, whereas the ONCVPSP pseudopotentials were employed instead [45]. Both sets of pseudopotentials showcased negligible deviation with respect to total energies. We used the Godby–Needs plasmon-pole model and a cutoff energy of 82 eV to describe the dielectric matrix. In order to assure convergence of the GW exchange-correlation self-energy, 2600 bands were employed, corresponding to an energy cutoff of 30 eV.

The thermodynamic Charge Transition Levels (CTLs) are defined as the Fermi energy at which two defect charge states have the same formation energy [46]. In simpler terms, a CTL represents the energy necessary for a transition to take place between two different charge states of a defect. In the particular case of substitutional nitrogen, such a transition can be expressed as,
(1)$$N_{\mathrm{Si}}^{q}⇌N_{\mathrm{Si}}^{q^{\prime }}\pm e^{-},$$
where $N_{\mathrm{Si}}^{q}$ and $N_{\mathrm{Si}}^{q^{\prime }}$ represent the defect at charge states *q* and $q^{\prime }$. If we consider the formation energy $E_{f}^{q}$ of the charge state $N_{\mathrm{Si}}^{q}$ as the reference, the formation energy of the charge state $N_{\mathrm{Si}}^{q^{\prime }}$ can be expressed as,
(2)$$E_{f}^{q^{\prime }}=E_{f}^{q}\pm \Delta \left(q/q^{\prime }\right)\pm \mu_{e}$$
In this equation, $\Delta (q/q^{\prime })$ represents the energy exchanged during the transition between charge states *q* and $q^{\prime }$. This difference accounts for the energy required to add or remove an electron from $N_{\mathrm{Si}}^{q}$, along with the energy released due to atomic relaxation. The chemical potential of electrons, $\mu_{e}$, is often written as the Fermi energy referred to the valence band maximum (VBM), $\mu_{e}=\epsilon_{\mathrm{VBM}}+\varepsilon_{F}$, where $\varepsilon_{F}\in \left[0,\epsilon_{\mathrm{CBM}}-\epsilon_{\mathrm{VBM}}\right]$, where $\epsilon_{\mathrm{CBM}}$ is the conduction band minimum. The CTL is then simply,
(3)$$\varepsilon^{q/q^{\prime }}=\pm \Delta \left(q/q^{\prime }\right)-\epsilon_{\mathrm{VBM}}.$$
From Equation (Equation 3), we evaluated the CTLs using two different approaches: the standard PBE approximation and the so-called DFT and GW combined approach [47]. Within the first method, $\Delta (q/q^{\prime })$ is simply evaluated as the difference in DFT total energies between the two relaxed defect structures at charge states *q* and $q^{\prime }$,
(4)$$\Delta \left(q/q^{\prime }\right)=E_{\mathrm{DFT}}^{q}\left\{R^{q}\right\}-E_{\mathrm{DFT}}^{q^{\prime }}\left\{R^{q^{\prime }}\right\},$$
where $R^{q}$ denotes the ground state geometry at charge state *q*. The ionization potential of the VBM, $\epsilon_{\mathrm{VBM}}$, is approximated by the corresponding Kohn–Sham eigenvalue in pristine silicon. In order to circumvent the self-interaction contribution in standard PBE calculations, the second approach relies on the GW method to determine accurate ionization potentials (IPs) and electronic affinities (EAs) [48,49]. The difference $\Delta (q/q^{\prime })$ is thus determined as an IP/EA, evaluated as a quasiparticle (QP) energy at charge state *q*, minus/plus a *relaxation energy*. This relaxation energy is evaluated as a DFT total energy difference between the relaxed structures at *q* and $q^{\prime }$, evaluated for the final $q^{\prime }$ charge state. The $\Delta (q/q^{\prime })$ difference is thus evaluated as,
(5)$$\Delta \left(q/q^{\prime }\right)=\mathrm{EA}/{\mathrm{IP}}^{q}\pm \left[E_{\mathrm{DFT}}^{q^{\prime }}\left\{R^{q^{\prime }}\right\}-E_{\mathrm{DFT}}^{q^{\prime }}\left\{R^{q}\right\}\right].$$
Finally, $\epsilon_{\mathrm{VBM}}$ in Equation (Equation 3) is evaluated as a QP energy of pristine Si.

The formation energy of charged defects (Equation (Equation 2)), is often corrected to mitigate the spurious electrostatic interaction between defect replicas [50]. In the present work, the correction is evaluated by aligning the Density Of States (DOS) of the charged and neutral defect states. In the case of the positive charge state, this shift is equal to −0.03 eV, whereas for both singly and doubly negative charge states it is equal to 0.01 eV.

## 3. Results and Discussion

### 3.1. The Symmetry Breaking in N${}_{\mathrm{Si}}$

Our first-principle calculations confirmed that the ground state geometry of N${}_{\mathrm{Si}}^{0}$ displays an off-center configuration with $C_{3v}$ symmetry, rather than the on-center configuration or the tetrahedral ($T_{d}$) symmetry. In the on-center configuration, four equivalent N-Si interatomic distances of 2.05 Å are found between the N atom and its nearest neighbours (or silicon atoms 1, 2, 3 and 4 in Figure 1). These Si atoms form a tetrahedron with an edge length of 3.34 Å. With the atomic distortion, the N atom moves away from one of its nearest neighbours (silicon atom 4 in Figure 1), elongating the N-Si distance to 3.15 Å (Figure 2a). The remaining three equivalent N-Si distances are shortened to 1.86 Å, similar to the N-Si bond length (1.73 Å) in silicon nitride [51]. The displacement of the N atom along the $C_{3}$ symmetry axis also distorts the tetrahedron formed by its nearest neighbours. The Si atom along the axis of distortion (denoted from now on the Si${}_{\langle 111\rangle }$ atom) is 3.87 Å away from the remaining three Si atoms, whose interatomic distances are 3.20 Å. Our analysis of the symmetry breaking indicates that the total displacement of the nitrogen atom away from its substitutional site is 0.57 Å. In its off-center configuration, the N atom lies relatively close (0.25 Å) to the interatomic plane defined by its nearest neighbours (or silicon atoms 1, 2 and 3 in Figure 1). Both these distances are in agreement with previously reported values [30,31].

To gain a deeper understanding of the symmetry breaking (SB) in substitutional nitrogen, we examined the one-electron defect states of both the on-center and off-center configurations. We approximated the one-electron wavefunctions using Kohn–Sham eigenfunctions and their corresponding energy levels using quasiparticle (Kohn–Sham) eigenvalues computed at Γ. Figure 2b shows the one-electron energies corresponding to the on-center and off-center geometries, along with the real space projection of the wavefunctions of the singlet defect states. In the on-center configuration, we identified a partially occupied $a_{1}$ state at 0.43 (0.20) eV from the valence band maximum (VBM). As depicted in Figure 2b, the one-electron wavefunction is centered in the N atom and is equally distributed among the four neighboring Si atoms. The unoccupied triplet states $t_{2}$, predicted in the theoretical model [24] (Figure 1), are found within the conduction band, at 0.09 (0.08) eV from the conduction band minimum (CBM). After the symmetry breaking, the partially occupied $a_{1}$ state lowers its energy, whereas the $t_{2}$ state splits into a second singlet $a_{1}$ and a doublet *e* (in agreement with symmetry considerations [52] and the previous model [24]). The partially occupied singlet state is located within the band gap, at 0.07 (0.02) eV from the VBM, and is primarily localized on the Si${}_{\langle 111\rangle }$ atom and to a lesser extent on the nitrogen atom. The second singlet state is found within the conduction band, at 0.03 (0.03) eV from the CBM. The doublet state hybridizes with the conduction states and is found at 0.11 (0.11) eV from the CBM. Notice that both the KS wavefunctions and the QP eigenvalues are computed at Γ and represented within the folded bulk bands. When k-points are added to the calculation, we, however, observe a large dispersion of the defect states that is more prominent for the $a_{1}$ state of the on-center configuration (for both spin channels) and the empty $a_{1}$ state of the off-center configuration (minority spin channel). The band structures for both defect configurations are shown in the Appendix A.

Based on the energy diagrams shown in Figure 2b and, in particular, the absence of partially occupied degenerate states, we confirm that the symmetry breaking $T_{d}\to C_{3v}$ is not a Jahn–Teller effect [22]. Thus, the PES of substitutional nitrogen does not exhibit the standard conical intersection, where the highly symmetric configuration is located at the top of the so called Mexican hat-shaped PES. In this case (or in the case of a pseudo Jahn–Teller effect [24]), the shape of the PES cannot be induced from symmetry considerations and so the nature of the critical points ($T_{d}$, $C_{3v}$) of the PES cannot be predicted without explicit electronic structure calculations. To begin exploring the PES of substitutional nitrogen in silicon, we examined the Minimum Energy Path (MEP) between the on-center and off-center configurations. In Figure 2c, the total DFT energy is plotted along the MEP, whose coordination reaction is taken as the displacement of the N atom away from its ideal on-center position. As depicted in Figure 2c, the MEP between these geometries corresponds to a linear interpolation between both structures. It should be noted that the energy difference between the two configurations is not negligible, with a value of 81 meV. This value is in decent agreement with the previous estimation of 50 meV [30] and 80 meV [29]. Additionally, we observe an energy barrier of 129 meV between the on-center and off-center configurations. Our results therefore indicate that both the tetrahedral ($T_{d}$) and the trigonal ($C_{3v}$) configurations are metastable.

We further explored the PES of substitutional nitrogen by investigating the transition path between equivalent off-center configurations. Since the symmetry breaking preserves any of the original $C_{3}$ symmetry axes, the nitrogen atom can move along any of the equivalent 〈111〉 directions away from its first silicon neighbors (Figure 1). As a result, the PES is characterised by four equivalent global minima as off-center configurations and the defect can reorient at finite temperatures. As a first approximation to the MEP between equivalent configurations, we tested a linearly interpolated path along which the N atom avoids its on-center site. However, the transition path relaxes to pass through the tetrahedral ($T_{d}$) symmetry, resulting in the MEP in Figure 2c. The N atom is therefore forced to transit through its on-center configuration to jump between equivalent off-center configurations. It is important to note that the activation energy for this reorientation mechanism cannot be simply estimated as the energy barrier between the on-center and off-center configurations, since the $T_{d}$ symmetry is a metastable minimum. Namely, the defect could remain trapped in its on-center configuration, delaying the transition to another global minimum. Before comparing our shape of the PES to experimental observations, we address the potential influence of size effects on the MEP. For this purpose, we performed relaxation calculations for both on-center and off-center defect structures within a 512-atom cell. The resulting interatomic distances differ by no more than 0.03 Å  from the previously described bond lengths. The energy difference between these structures is 81 meV and the energy barrier is 129 meV, indicating that the MEP obtained with the 216-atom cell converged within 10 meV.

### 3.2. Ab Initio EPR Parameters

Once the ground state configuration of the neutral substitutional nitrogen is well described, we calculate its corresponding EPR parameters to provide an explicit comparison with experimental data. In particular, in Table 1, we show, for the off-center configuration, the principal values of the $\overset{↔}{g}$ tensor and of the hyperfine tensors together with their isovalues ($g_{\mathrm{iso}}$ and $A_{\mathrm{iso}}$). The hyperfine tensors refer to the silicon atom along the atomic distortion (or the Si${}_{\langle 111\rangle }$ atom) and to the nitrogen impurity. The substantial disparity between $A_{i}$ principal values corresponding to the Si${}_{\langle 111\rangle }$ atom and the N atom reflects the notable localization of the unpaired electron on the Si atom (see wavefunction of the $a_{1}$ state in Figure 2b).

Concerning the $\overset{↔}{g}$ tensor, we account for the axial symmetry of the defect and we designate the principal value $g_{1}$ as the one corresponding to the principal axis parallel to the 〈111〉 direction (or $g_{\|}$). Consequently, the $g_{2}$ and $g_{3}$ values correspond to any two axes perpendicular to the 〈111〉 direction (or $g_{⊥}$). From our values in Table 1, it is apparent that the axial symmetry is not fully preserved, as indicated by the inequality of $g_{2}$ and $g_{3}$. This small deviation is within the expected inaccuracy estimates of the GIPAW method [53,54] and cannot be attributed to a poor description of the defect geometry, since the estimated interatomic distances are fully compliant with the $C_{3v}$ symmetry. However, it can be assigned to the artificial dispersion of the defect states when several *k*-points are included in the sampling of the Brillouin zone (see band structures in the Appendix A).

In Table 1, we compare the EPR parameters calculated for the off-center configuration to the reported EPR data of the SL5 center [18,19,20]. The calculated principal values of the hyperfine structure tensor $\overset{↔}{A}$ are within 12% of the experimental ones for the silicon atom along the 〈111〉 distortion and within 20% for the nitrogen impurity [18,19,20]. Moreover, both the principal values and the isovalue of the $\overset{↔}{g}$ tensor lie within ∼1000 ppm of those measured via Brower [18,19] and with even better agreement (∼700 ppm) with the most recent measurements [20].

Earlier theoretical calculations of the hyperfine tensors, for the off-center configuration, were derived using the Hartree–Fock approximation for a relatively small atomic cluster NSi${}_{4}$H${}_{12}$ [55]. However, the calculated isovalues exhibited significant discrepancies when compared to experimental EPR data [18,19,20]. Specifically, the calculated value for $A_{\mathrm{iso}}{(}^{29}\mathrm{Si})$ was overestimated, while the one for $A_{\mathrm{iso}}\left(N\right)$ was underestimated, with deviations of nearly a factor of 2 in both cases [55]. By contrast, our ab initio EPR parameters (both $\overset{↔}{g}$ and $\overset{↔}{A}$ tensors) display excellent agreement with the experimental data and hence provide a firmly grounded theoretical assignment of the SL5 EPR center to the distorted off-center configuration of the substitutional nitrogen.

Besides investigating the magnetic properties of a frozen N atom, the dynamics of the SL5 center were also investigated from EPR stress measurements [18,19]. In particular, the broadening of the spectral lines at temperatures above 35 K was attributed to the reorientation of the defect between equivalent off-center configurations. Based on this analysis, the activation energy for the N atom to jump from one 〈111〉 distortion to another was inferred to be 0.11 ± 0.02 eV [19]. From our exploration of the PES, we calculate an energy barrier of at least 0.129 eV between equivalent off-center distortions (Figure 2c). This evaluation is in rather good agreement with the experimental estimate of 0.11 eV. Notice, however, that given the existence of a metastable on-center minimum in our reorientation path, the effective activation energy for the reorientation process is expected to be slightly higher. Before even estimating this effective barrier, in the following, we reassess the existence of a metastable on-center configuration.

From EPR measurements [18,19,20,21,25,26], an argument in favor of a metastable on-center configuration was put forward to explain the observed increase in the hyperfine splitting when augmenting the temperature [21,26]. In order to elucidate the existence of this metastable minimum, we estimated its EPR parameters (Table 1) and compared these values to the ones previously assigned to this tetrahedral configuration [21,26]. In this highly symmetric configuration, the $\overset{↔}{g}$ tensor and the $\overset{↔}{A}$(N) hyperfine tensor are expected to be isotropic. In fact, as reported in Table 1, we found an isotropic hyperfine tensor with a considerably large isovalue (165.7 MHz), while the isovalue on the silicon ($|A_{\mathrm{iso}}{(}^{29}\mathrm{Si})|=40.1$ MHz) is about the same as found for the nitrogen impurity in the off-center configuration. Because of numerical instabilities in the calculations of the $\overset{↔}{g}$ tensor for the on-center configuration, we performed an EPR demonstration calculation on a hydrogen-passivated silicon cluster; specifically, the NSi${}_{4}$H${}_{12}$ cluster. This cluster features four N-Si bonds with a length of 1.89 Å and Si-Si distances of 3.08 Å. As shown in Table 1, both the calculated $\overset{↔}{g}$ and $\overset{↔}{A}$(N) tensors are isotropic and the value of the hyperfine coupling constant (226.6 MHz) is consistent with the one (165.7 MHz) calculated for the on-center N${}_{\mathrm{Si}}$ in the silicon supercell.

The calculated hyperfine tensors $\overset{↔}{A}$(N) of the on-center configuration are in disagreement with the experimental room temperature hyperfine splitting value given in [21]. Yet, we note that in [26] the temperature dependence of the hyperfine splittings is fitted by means of Boltzmann weights. From the fit, a remarkably large value of 376 MHz was assigned to the N in the on-center configuration. From our model, the defect state is transferred from N to Si when moving from on-center to off-center configuration (see Figure 2c). This implies that the value of $A_{\mathrm{iso}}$(N) drastically decreases with this transfer; thus, it would seem unlikely that this value only differs by about 5 MHz between the two configurations [19,21,28]. A remarkably larger difference can be expected, as suggested by Murakami’s fit [26] and by our results presented in Table 1. In fact, $A_{\mathrm{iso}}$ should be significantly larger for the atom around which the paramagnetic state is located. In our calculations (Table 1), we see, for instance, a $A_{\mathrm{iso}}{(}^{14}N)$ and a $A_{\mathrm{iso}}{(}^{29}\mathrm{Si})$ of 226.6 MHz and −28.3 MHz, respectively, for the on-center configuration that become 32.2 MHz and −264.1 MHz, respectively, in the off-center configuration.

From our computational model, we remark that even if a metastable on-center minimum is found in the exploration of the PES, the ratio between the off-center and on-center occupation (from Boltzmann weights) is about 120 at room temperature. Hence, its implication in the thermally activated dynamics is minimal and its contribution to the *averaged* spectroscopic signals is minor, leading just to a small increase (∼5 MHz) in the hyperfine splittings at room temperature. Yet, the on-center configuration can be tracked down from the temperature-induced increase in the hyperfine splittings [26]. The latter, however, might also involve an effect due to coupling with the phonons of the off-center potential well.

We note that, contrary to the interpretation suggested by Ref [21], the on-center configuration should not be mistaken with the room- or high-temperature averaged configuration of N${}_{\mathrm{Si}}$. In fact, the averaged spectroscopic signals depend on the occupation probability of both minima (the four degenerate off-center and the on-center).

### 3.3. Charge Transition Levels of N${}_{\mathrm{Si}}$

Once a computational model for neutral substitutional nitrogen has been proposed, we investigate its relative stability compared to other charge states. In particular, we investigate the charge states +1, −1 and −2 and we determine their respective CTLs. At charge state +1, there is no *trapped* electron within the silicon band gap and the defect remains in an on-center configuration, with a N-Si bond length of 2.05 Å. At charge state −1, the off-center configuration is preferred and so the $a_{1}$ defect state is fully occupied. Along the distortion, the N-Si${}_{\langle 111\rangle }$ bond is 3.07 Å, whereas the remaining three N-Si distances are equal to 1.85 Å. If yet another electron is added to the defect, the system remains in an off-center configuration, with a characteristic N-Si${}_{\langle 111\rangle }$ distance of 1.85 Å, compared to the three N-Si bond lengths of 3.04 Å. This charge state −2 is paramagnetic, with the defect state $a_{1}$ in the conduction band partially occupied.

We present our calculated thermodynamic CTLs, obtained within the PBE+GW (PBE) approach, for the single donor and the single acceptor levels of substitutional nitrogen in Table 2. The donor level is found relatively close to the valence band maximum, with calculated values of $\epsilon_{\mathrm{VBM}}+0.19\left(0.08\right)$ eV. In contrast, the acceptor level is located deep in the silicon band gap, positioned at $\epsilon_{\mathrm{CBM}}-0.55\left(0.32\right)$ eV from the conduction band minimum. Notice that both values correspond to the mid-region of the silicon band gap, considering that the band gap is estimated to be 1.02 (0.63) eV. Finally, the double acceptor level is found within the conduction band, at 0.06 (0.06) eV from the conduction band minimum. The charge state −2 is therefore thermodynamically unstable.

The existence of a single donor and acceptor levels for N${}_{\mathrm{Si}}$ was previously reported via an ab initio calculation [29]. The proposed values were $\epsilon_{\mathrm{VBM}}+0.5$ eV and $\epsilon_{\mathrm{CBM}}-0.4$ eV, respectively. Notice, however, that they reported an electrical gap of 0.5 eV and that a corrective shift of the estimated levels was made. Contrary to our approaches (Table 2), their corrected levels are both deep in the band gap, at ∼0.25 eV from each other. Given the well-established accuracy of the DFT + GW approach estimating single donor and acceptor levels, we emphasize the existence of a donor level close to the valence band maximum.

In Table 2, we present a comparison between our CTLs and the DLTS activation energies tentatively assigned to the off-center substitutional nitrogen [17,28]. We observe a significant disparity between the reported values in [28] and our estimated levels. Given the accuracy of the DFT + GW method [47], we argue that the assignment of the levels $\epsilon_{\mathrm{CBM}}-0.31$ eV and $\epsilon_{\mathrm{CBM}}-0.08$ to substitutional nitrogen was incorrect. In contrast, we notice the proximity of our estimated acceptor level with the one proposed in a recent DLTS investigation [17]. Nonetheless, we acknowledge the discrepancy between their identified double acceptor level and our computed value. From this disparity, we conclude that either the assignment of both acceptor levels to substitutional nitrogen was incorrect or that these levels are the signature of another N-related defect (e.g., the nitrogen–vacancy complex).

## 4. Conclusions

We performed a systematic investigation of the symmetry-breaking mechanism in substitutional nitrogen from first-principle calculations. We characterized fundamental properties of both the on-center and off-center configurations of N${}_{\mathrm{Si}}$, including the relative position of its single-particle defect states within the band gap. Moreover, we proposed a model for its potential energy surface by performing an extensive search of the minimum energy path between equivalent off-center configurations. Such a transition path forces the nitrogen center to pass through the on-center configuration that is estimated to be a shallow metastable minimum.

In order to provide a direct comparison with spectroscopic signals, we computed the EPR parameters and the thermodynamic CTLs of N${}_{\mathrm{Si}}$. Our EPR values for the off-center configuration are in excellent agreement with the SL5 EPR signature. Moreover, the energy barrier we calculate for the reorientation mechanism between equivalent off-center configurations is in fair agreement with EPR stress measurements, validating our model for its potential energy surface. Furthermore, our calculations further support a large hyperfine splitting value for the N${}_{\mathrm{Si}}$ in the on-center configuration as estimated in [26]. Finally, we proposed a set of deep levels for N${}_{\mathrm{Si}}$, corresponding to a single donor and acceptor levels. Our values allowed us to identify incorrect assignments between DLTS activation energies and the electrical activity of N${}_{\mathrm{Si}}$ and they can guide its identification in future works.

## Figures and Tables

**Figure 1 nanomaterials-13-02123-f001:**
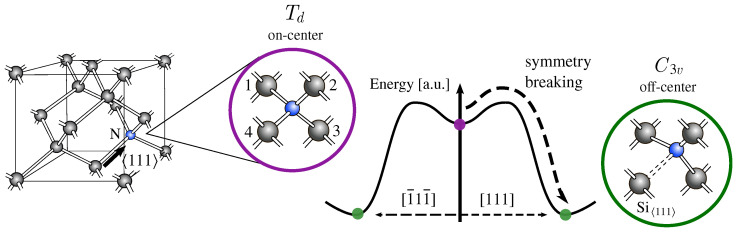
Substitutional nitrogen in silicon. The proposed symmetry-breaking (SB) mechanism $T_{d}\to C_{3v}$ for this point-defect is depicted for nitrogen (in blue) and its first silicon neighbours. In the lower $C_{3v}$ symmetry, the Si atom positioned along the axes of the distortion or the 〈111〉 direction is explicitly labelled. The suggested PES corresponding to this SB is also illustrated. Two equivalent off-center minima with $C_{3v}$ symmetry are represented in green for the crystal directions $\left[\bar{1}1\bar{1}\right]$ and [111]. A metastable on-center minimum ($T_{d}$ symmetry) is depicted in purple.

**Figure 2 nanomaterials-13-02123-f002:**
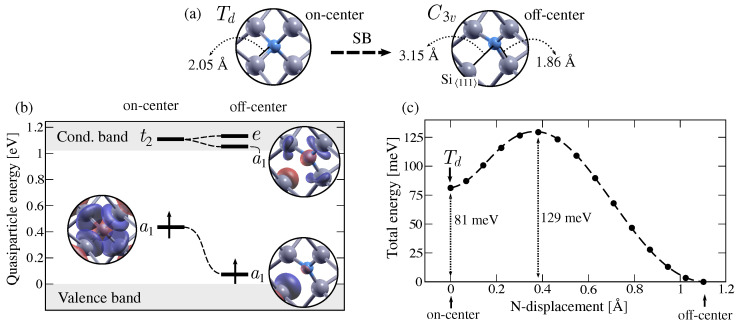
Ab initio characterization of the symmetry-breaking mechanism (SB) in substitutional nitrogen. (**a**) Relaxed defect structures depicting nitrogen and its nearest neighbours for the on-center and off-center configurations. The interatomic distances between nitrogen and its first neighbours are displayed. (**b**) Defect-induced states for the on-center and off-center configurations. Quasiparticle (QP) energies of defect states estimated at Γ are represented with respect to the valence band maximum and the conduction band minimum for the spin-up channel. Defect states are labelled according to their irreducible representation and the Kohn–Sham wavefunctions corresponding to singlet states $a_{1}$ are shown. (**c**) Minimum energy path (MEP) between the on-center and off-center configurations. The DFT total energy is plotted with respect to the displacement of the N atom from its on-center position to its off-center configuration. The energy difference between the two minima is shown, as well as the energy barrier from the off-center to the on-center geometries.

**Table 1 nanomaterials-13-02123-t001:** EPR parameters of substitutional nitrogen in silicon. For the off-center configuration, the estimated $\underset{g}{↔}$ and the hyperfine $\underset{A}{↔}$ tensors for the N impurity (${}^{14}$N) and Si atom along the 〈111〉 distortion (${}^{29}$Si${}_{\langle 111\rangle }$) are shown (tw-off), together with EPR data of the SL5 center [18,19,20]. Calculated hyperfine $\underset{A}{↔}$ tensors are also displayed for the on-center geometry in the 216-atom cell (tw-on) and in the NSi${}_{4}$H${}_{12}$ cluster (tw-cluster). These values are compared to EPR data [21,26]. Also, the $\underset{g}{↔}$ tensor as calculated for the NSi${}_{4}$H${}_{12}$ cluster is given.

	$\overset{↔}{g}$				$\overset{↔}{A}$ (${}^{29}$Si${}_{\langle 111\rangle }$) (MHz)				$\overset{↔}{A}$ (${}^{14}$N) (MHz)			
Ref.	$g_{1}$	$g_{2}$	$g_{3}$	$g_{\mathrm{iso}}$	$A_{1}$	$A_{2}$	$A_{3}$	$A_{\mathrm{iso}}$	$A_{1}$	$A_{2}$	$A_{3}$	$A_{\mathrm{iso}}$
tw-off	2.0020	2.0089	2.0078	2.0062	−380.4	−205.9	−205.9	−264.1	37.5	29.6	29.6	32.2
[18,19]	2.0026	2.0089	2.0089	2.0068	397.5	231.7	231.7	287.0	45.3	36.3	36.3	39.3
[20]	2.00219	2.00847	2.00847	2.00638	396.6	234.7	234.7	288.7	45.85	36.49	36.49	39.61
tw-on	-	-	-	-	−55.0	−32.6	−32.6	−40.1	165.7	165.7	165.7	165.7
tw-cluster	2.0056	2.0056	2.0056	2.0056	−58.4	−13.2	−13.2	−28.3	226.6	226.6	226.6	226.6
[21] ([26])	-	-	-	2.0065	-	-	-	-	-	-	-	45.5 (376.2)

**Table 2 nanomaterials-13-02123-t002:** Deep levels located within the bandgap associated with substitutional nitrogen in silicon. Thermodynamic CTLs were calculated within a standard PBE approximation (tw-PBE) and within the combined DFT and GW approach (tw-PBE+GW). Previously estimated levels (pw) are also included [29]. DLTS levels were tentatively assigned to susbstitutional nitrogen [28] and activation energies associated with the off-center nitrogen, however, also addressed as NV center, in [17]. All levels are given from the conduction band minimum in eV.

Reference	ϵ(+/0)	ϵ(0/−)	ϵ(−/2−)
tw-PBE	0.55	0.32	-
tw-PBE + GW	0.83	0.55	-
pw [29]	-	0.40	-
Exp. [28]	0.31	0.08	-
Exp. [17]	-	0.64	0.34

## Data Availability

The input and output files of all calculations are available upon request.

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
