# Peer review of "Deep Levels and Electron Paramagnetic Resonance Parameters of Substitutional Nitrogen in Silicon from First Principles"

_nanomaterials, 2023, doi:10.3390/nano13142123_

Round 1

Reviewer 1 Report

The presented  paper reports on advanced solid-state quantum simulations aimed at providing insights on formation and electronic properties of well-known substitutional nitrogen defect in silicon, in both stable and metastable configurations. Introduction provides detailed historical overview on NSi defect studies. The paper is well-written, the results are mostly presented in a clear manner.  However, results on charge transition levels do not look convincing, that part needs a rework.

1) The symbols for g-tensor and A hyperfine coupling constant. It is not clear why they are with double arrow on top, none of the cited papers use such notation.

2) You mention -2 charge state, but then do not provide any information about it: geometry, electron localization and CTLs in Table 2. You should either add something about it or remove any mentions of it.

3) CLT part and table 2. There is much more experimental measurements on nitrogen in silicon: https://iopscience.iop.org/article/10.1143/JJAP.21.L443/pdf
https://journals.aps.org/prb/pdf/10.1103/PhysRevB.10.638

It is not clear why both PBE and PBE+GW are presented in the table, when you mention "Given the accuracy of the DFT+GW method, we argue... " So the PBE itself is wrong? With an underestimated band gap of 0.63 eV is it reasonable to calculate EC - x values?

In Ref. 7 there is a sentence: "... off-center substitutional nitrogen with an optical deep donor level at EC − 0.58 eV13 and a thermal donor level at EC − 0.33 eV.11,12"

Why does these values do not appear in Table 2?

It is also worth mentioning which, optical or thermal, charge transitions are calculated. For optical (vertical) additional correction may be required (DOI: 10.1103/PhysRevB.101.020102)

Reviewer 2 Report

The authors of the manuscript provided solid investigation of Nitrogen defects in Silicon by theoretical methods. The manuscript is well written and well organized and can, in principle, published at present form.

I recommend just non-significant improvements:

- line 245. The authors discuss difference between g2 and g3 values and assigned the difference to some physical aspects. However, in my opinion, this small difference between g2 and g3 is due to inaccuracy of GIPAW method implemented in Quantum Espresso. The estimate of the inaccuracy in GIPAW calculations was made in J. Appl. Phys. 125, 185701 (2019) https://doi.org/10.1063/1.5092626  

- it will be better to replace in the whole manuscript the phrases "bottom of the conduction band (BCB)" and "top of the valence band (TVB)" to widely accepted "conduction band minimum (CBM)" and "valence band maximum (VBM)" 

- line 25. If it possible, please provide the reference to this statement

- line 50. You are writing about some energy barrier but did not provide the value

- line 169. The abbreviation SB is defined second time

Round 2

Reviewer 1 Report

Authors provided answers to all the comments. The paper now can be accepted in present form.